# Sympathy-Empathy and the Radicalization of Young People

**DOI:** 10.3390/children9121889

**Published:** 2022-12-02

**Authors:** Nathalie Lavenne-Collot, Nolwenn Dissaux, Nicolas Campelo, Charlotte Villalon, Guillaume Bronsard, Michel Botbol, David Cohen

**Affiliations:** 1Service de Psychiatrie de l’Enfant et de l’Adolescent, Centre Hospitalier Universitaire de Brest, 29200 Brest, France; 2Faculté de Médecine, Université de Bretagne Occidentale, 29200 Brest, France; 3Laboratoire du Traitement de l’Information Médicale, Inserm U1101, 29200 Brest, France; 4Service de Psychiatrie de l’Enfant et de l’Adolescent, Assistance Publique-Hôpitaux de Paris, Hôpital Pitié-Salpêtrière, 75006 Paris, France; 5Laboratoire de Psychologie Clinique, Psychopathologie, Psychanalyse (EA 4056), Institut de Psychologie, Université Paris Descartes, Sorbonne Paris Cité, 92100 Boulogne-Billancourt, France; 6Département de Sciences Humaines et Sociales, Laboratoire Soins Primaires, Santé Publique, Registre des Cancers de Bretagne Occidentale (EA 7479), 29200 Brest, France; 7Laboratoire de Santé Publique (EA3279), Aix Marseille Université, 13005 Marseille, France; 8Professeur Émérite de Psychiatrie de l’Enfant et de l’Adolescent, Université de Bretagne Occidentale, 29000 Brest, France; 9CNRS, UMR 7222, Institut des Systèmes Intelligents et de Robotiques, Sorbonne Université, 75006 Paris, France

**Keywords:** radicalization, empathy, callous unemotional traits, adolescence, violent extremism

## Abstract

Background: The sympathy-empathy (SE) system is commonly considered a key faculty implied in prosocial behaviors, and SE deficits (also called callous-unemotional traits, CUTs) are associated with nonprosocial and even violent behaviors. Thus, the first intuitive considerations considered a lack of SE among young people who undergo radicalization. Yet, their identification with a cause, their underlying feelings of injustice and grievance, and the other ways in which they may help communities, suggest that they may actually have a lot of empathy, even an excess of it. As a consequence, the links between SE and radicalization remain to be specified. This critical review aims to discuss whether and how SE is associated with developmental trajectories that lead young people to radicalization. Method: We first recall the most recent findings about SE development, based on an interdisciplinary perspective informed by social neuroscience. Then, we review sociological and psychological studies that address radicalization. We will critically examine the intersections between SE and radicalization, including neuroscientific bases and anthropologic modulation of SE by social factors involved in radicalization. Results: This critical review indicates that the SE model should clearly distinguish between sympathy and empathy within the SE system. Using this model, we identified three possible trajectories in young radicalized individuals. In individuals with SE deficit, the legitimization of violence is enough to engage in radicalization. Concerning individuals with normal SE, we hypothesize two trajectories. First, based on SE inhibition/desensitization, individuals can temporarily join youths who lack empathy. Second, based on an SE dissociation, combining emotional sympathy increases for the in-group and cognitive empathy decreases toward the out-group. Conclusions: While confirming that a lack of empathy can favor radicalization, the counterintuitive hypothesis of a favorable SE development trajectory also needs to be considered to better specify the cognitive and affective aspects of this complex phenomenon.

## 1. Introduction

Radicalization is a complex phenomenon representing a significant threat worldwide. Young people, from the age of 12/13, have also been concerned, thus manifesting specific adolescent issues that have not been studied as much as other forms of juvenile violence [1]. While the growing threat of Right-Wing Extremism in some countries is associated with a tendency to recruit older rather than younger people, Islamic radicalization of young people remains an ongoing threat and a social challenge.

In addition, the radicalization epidemic that occurred in European countries from 2014/2015 among young people who converted to Islam without Muslim backgrounds has raised unsolved issues: Why does someone radicalize against his own people? What type of empathy do radicalized people feel for someone toward whom they direct their radicalized violence, given that the development of the sympathy-empathy (SE) system is particularly activated during adolescence and young adulthood [2,3]?

The SE system is commonly involved in prosocial behavior, and SE deficits are associated with many conditions involving antisocial and violent behavior. An SE deficit is notably one of the hallmarks of psychopaths [4] and a subgroup of adolescents with severe conduct disorders and callous unemotional traits (CUTs) [5]. Terrorists do indeed kill civilians in large numbers, suggesting an obvious lack of empathy. However, the recent implications of ‘ordinary adolescents’ without any psychiatric disorder put this hypothesis into perspective. Moreover, their identification with a cause, the underlying feelings of injustice and grievance that motivate their radicalization, and the other ways in which they may help communities, suggest that they may instead demonstrate preserved empathetic abilities. Indeed, several case studies and surveys have shown that at least some young people actively involved in radicalized violent behaviors, especially suicide bombing, have been strongly committed to volunteering and philanthropic engagement, thus suggesting empathetic capacities that might be higher than those of the usual offenders [6,7,8]. Moreover, the apparent contradiction between preserved empathy and violent behaviors has already been noticed among adults in other circumstances (e.g., Syndrome E [9]).

In an attempt to overcome this apparent contradiction and compensate for the lack of in-depth data about the role of the SE system in radicalization, this literature review aims to explore whether and how SEs are involved in the radicalization among youth from the first stage of enrollment to possible violent action. In doing so, we will propose that the above-mentioned contradiction may well have to do with SE itself, especially with the modulation of SE by several factors that may be involved in the radicalization of young individuals. Moreover, while the SE model has relevance regardless of age, it is particularly important to consider this with young people as they are progressing through important stages of development and socialization, and this enhances the potential for protective measures to have some concrete effect.

In Section 1, after an introduction to the SE system, we will examine its role in the sensitivity to the face-to-face encounters in group or virtual social media propaganda discourse; we will then consider both the influence of emotional contagion in increasing group identification and the factors involved in the SE-based motivational aspects of radical commitment. These findings will ground the question at the center of this paper detailed in Section 2: the consistency between empathy and radical views and between beliefs and behaviors and the type of empathy that may be involved in this consistency. We will distinguish, in addition to radicalized youth showing a deficit of empathy classically described in young offenders, other radicalized young people who might present a dissociated profile combining a higher affective component (including emotional) contrasting with a weaker cognitive component of empathy.

After a brief intermediary summary, Section 5 will examine how this dissociated profile may be triggered by the environmental context in which radicalization occurs; e.g., in certain contexts, the opposition between in-group and out-group relations may contribute to a dehumanization of the out-group, and this dehumanization may favor violent radicalized behaviors toward the out-group’s members. 

The last section (Section 4) will lead to one of the main contributions of this this work: the hypothesis of a so-called “double dissociation model of sympathy/empathy in youth radicalization”. This model of adolescent radicalization is summarized in Figure 1 from a micro-macro perspective. The first dissociation occurs at the individual level (micro) and is characterized by high affective SE contrasting with diminished cognitive empathy. The second occurs at the intergroup level (macro), thus combining high in-group empathy and low out-group empathy both inversely related according to a parochial empathy effect; i.e., an increase in one leads to a decrease in the other. The implications of these hypotheses will be discussed as well as the need for specific empirical research to confirm them.

## 2. Relationship between the Sympathy/Empathy System and Radicalization in Young People

### 2.1. The Sympathy/Empathy System

Empathy has a broad understanding in folk psychology and is influenced by information-processing biases, perspective tacking, theory of mind, and reasoning. Empathy is composed of at least a spontaneous transfer of emotion and a concern for others’ well-being and has several possible behavioral outcomes, such as altruism, compassion, kindness, liking, and trust [10,11]. Therefore, empathy is a multidimensional construct (with emotional, cognitive and motivational facets) reflecting an ability to feel and understand another person’s lived experience and associated mental state while mentally adopting that person’s perspective [12]. Empathy encompasses the automatic embodiment of internally feeling what another person is experiencing. Empathy implies emotional processing, a cognitive theory of mind, and self-regulation and relies on the interaction of interacting neural regions within topographically and functionally distinct networks [13]. From an evolutionary perspective, an empathetic response is composed of several layers that build upon each other and remain functionally integrated and related to different levels of empathy. At the core of the empathetic response lies emotional contagion (or sympathy), the middle layer includes empathetic concern, and the outer layer contains perspective-taking and targeted helping [14]. More importantly, this theorization supports a developmental perspective of empathy: while some empathic abilities are present from early infancy, others depend on higher cognitive functions that emerge later in development due to brain progressive maturation under the influence of environmental factors [15].

Empathy is involved in many aspects of emotional processing, notably through the mechanism of emotional sharing (also called emotional contagion or affective resonance or sympathy) [12]. The basic tenet of these models is that observing an action in another individual directly activates the matching neural substrates in the observer through which the action can be understood and leads the observer to vicariously experience similar feelings [16]. This state-matching reaction has been related to simulation theory relying on the mirror neuron system (MNS), which is the first primitive resonance mechanism involved in empathy processes since early infancy [17]. Individual differences in empathy, notably among subjects who score higher on a measure of empathy, activate the subjects’ MNS more strongly when the subjects hear about the actions of others [18].

Indeed, emotional contagion refers to sympathy. However, empathy requires the ability to distinguish between simultaneous representations of both the other’s and one’s own current experiences or feelings [19]. Maintaining a self/other distinction (SOD) is a fundamental prerequisite also supported by neuroimagery data showing that empathy does not involve a complete self-other merging [20]. Notably, SOD permits one to shift from emotional contagion to empathy by distinguishing that the primary source of one’s feeling is the perception of someone else’s experience. Another important difference relies on an own-body-transformation (OBT) that helps one decenter from oneself into the other’s body and mind within a shift from an ego-centered (sympathy) to heterocentered (empathy) perspective [21]. Previous research has demonstrated the existence of separate brain networks recruited from egocentric and allocentric perspectives [22] and decoupling mechanisms between self- and other-centered processes [23]. If perspective-taking also involves executive functions (e.g., inhibition [24]), SOD must be conceived as an independent mechanism that relies on others’ brain networks and is related to self-consciousness and awareness in its different dimensions [25]. The best evidence of the essential place of self-awareness in empathy comes from Bischof, who found time synchronicity in the onset of mirror recognition and empathic behaviors for peers during early development [26].

Thus, although empathy is a complex and multifaceted concept [27], it is legitimate to distinguish, as modern neuroscience does, sympathy (affective sharing/emotional contagion) from empathy (a concept that requires being in the other’s shoes).

### 2.2. Sympathy/Empathy and Sensitivity to Propaganda

Youth radicalization occurs through both face-to-face group encounters and internet and virtual contacts and is more likely to result when these two processes happen simultaneously [28,29,30]. Despite their important role in affecting recipients’ beliefs, attitudes, and intentions, little is known about individual dispositional factors involved in sensitivity to propaganda whether online or offline [31]. We propose to explore here to what extent empathic abilities may be involved at this early stage of radicalization.

First, SE is involved in the way one reacts to media viewing. SE has been associated with (i) increased physiological activation when one views movie clips containing graphic scenes of horror [32]; (ii) coping strategies [33]; (iii) adoption of altruistic behaviors [34]; and (iv) perception of a higher level of danger severity and risk for oneself when one views victimization stories [35]. One of the most powerful communication tools used by terrorist groups is narratives [36]. The crucial role of narratives in radicalizing adolescents is supported by numerous authors who have highlighted the fundamental place of narratives in building an attractive epic or in mythological storytelling [28,37,38,39,40]. The powerful effect of narrative propaganda relies on eliciting affective reactions in the reader [41], and empathy is involved in many aspects of emotional processing, notably through the mechanism of emotional sharing (also called emotional contagion or affective resonance or sympathy) [12].

The main themes and emotions elicited in a propaganda narrative are scenes of victimization (for instance, animal experimentation) that may arouse high empathetic abilities (such as empathetic concern and even compassion) and emotional responses (especially disgust, anger, sadness and guilt) [42,43]. Interestingly, most of these emotions have adaptive and survival values and thus elicit rapid nonconscious responses [44]. More importantly, emotional contagion involves many components: unconscious and automatic mimicry [45] and a conglomerate of somatosensory sensations that facilitate physiological and motor feedback inducing emotion in the receiver. For instance, observing and feeling disgust activate the same sites on the anterior insula and visceral sensations, such as nausea [46]. Disgust is particularly relevant since it is related to violations of internalized moral duties [47], frequently associated with the adoption of obsessive and compulsive behaviors [48] and with dehumanization [49], which occupies a prominent position in radicalization (see below). The limbic system plays an important role in this stage because amygdala activity has been shown to be higher in adolescents than in adults who view social threat-related stimuli [50]; therefore, this first emotional response may be increased.

Several arguments support the idea that emotional contagion or sympathy also has a central place far beyond recruitment [51]. Social psychology emphasizes the central role of emotional contagion in group cohesion [52]. Notably, sharing emotions is an important factor in the differentiation of friends from foes because of an in-group favoritism involved in emotion recognition [53,54]. Moreover, activation of the premotor cortex of the MNS when one observes an action performed by someone else is not neutral for the observer’s desires: this activation may increase the attractiveness of the goal pursued by the action. Such mimetic desire involves an influence of the MNS on the brain valuation system (BVS), thereby increasing the value of the goal targeted by the observed [55]. This phenomenon may be an effective way of acquiring new values and behaviors through a mechanism of non-verbal contagion [56].

Second, case reports of young people engaged in radicalization sometimes connect engagement to the repetitive viewing of propaganda movies [37,39]. Importantly, SE involves not only these first bottom-up primitive resonance mechanisms but also higher top-down abilities needed to moderate its effects [12]. Previous research notably showed that repetitive exposure to painful situations might lead to an inhibition of the representation of pain in the observer when painful procedures are inflicted [57]. Interestingly, such sympathy inhibition mechanisms may provide young people engaging in radicalization with the feeling of reaching a superhuman status [39,40,58].

Third, the persuasive effect of narratives relies on narrative transportation, i.e., the ability to be totally immersed in a story [59]. In particular, identification with protagonists of narratives mediates this effect: as readers mentally simulate the events that happen to a character, readers may come to understand what it is like to experience the described events [60]. As empathy perspective-taking is related to mental imagery and transformation aimed at the ability to feel like someone else, empathy perspective-taking should be involved in both narrative transportation and identification. Some studies indeed found a positive correlation between empathy and identification with the characteristics of a story [61]. Interestingly, positive associations were also found between empathy and “fan identity” in media viewing [62], whereas identification with a charismatic leader is supposed to be an important determinant in young radical commitment [63,64].

Recently, in an EEG study, Yoder et al. presented 238 adult participant video clips containing ISIS propaganda (either heroic or social martyr narratives) and collected behavioral measures of appeal, narrative transportation, sympathy/empathy and attraction to terrorism [31]. The findings confirmed that sympathy/empathy plays an important role in psychological predispositions to attraction by terrorism since individuals with higher dispositional empathy reported greater narrative transportation. Moreover, higher empathetic subjects preferred heroic narratives, thereby indicating that at least a subgroup of subjects with high empathy would be the intentional targets of narratives emphasizing individual benefits, notably personal glory and empowerment through sacrifice and righteous violence; this indication is stunning with regard to motivations commonly related to empathy.

### 2.3. Empathy/Sympathy and Radical Group Identification

In the previously mentioned EEG study exploring neural processes impacted by propaganda videos, heroic narratives were associated with electrophysiological patterns accompanying autobiographical remembering [31]. Autobiographical memory deals with identity [65], which is a central issue in adolescents’ motivation for radical commitment, according to uncertainty-identity theory [66,67]. Some clinical observations support this self-other identification in radicalized young people; for instance, an adolescent became engaged after viewing a propaganda video showing violence against women who looked like her own mother [63], or reactivating suffering linked to personal family history [64].

Since several authors do not distinguish between empathy (self-other distinction) and sympathy (self-other identification) [27], especially in the field of social psychology, it is rather perspective-taking that is conceived as activating the self-concept and leading perspective-takers to attribute a greater proportion of their self-traits to the other according to the self-other overlap hypothesis [68,69]. In the context of radicalization, some authors have emphasized this identity fusion by using a series of increasingly overlapping circles, one of which represented the self and the others a given group [70,71]. Moreover, we argue that several factors would favor a total loss of SOD (or complete self/other merging) in adolescents as compared to older subjects for the following reasons:(1)Developmental immaturity of the SE system with regard to the development of top-down regulation abilities. Several studies suggest that adolescents’ neural response patterns may differ from adults’ patterns in situations that evoke cognitive or emotional empathy [72]. The development of neural circuits underlying empathy from childhood (7 years) to adulthood (40 years) through fMRI starts with emotional empathy that appears earlier than cognitive empathy, but this development also shows a gradual shift from a visceral response to pain as a potential threat to a more detached and regulated appraisal of the stimulus [73]. Neuroimagery studies have shown the following: stronger automatic responses in adolescents who witness another in a painful situation [74]; a negative relationship between empathetic accuracy and brain activation (this relationship is compatible with adolescents becoming immersed in their own emotions while sharing the emotional experience of the target) [75]; and compensatory hyperactivation of emotionally related brain areas to compensate for adolescents’ lower emotional empathy ability [76].(2)SOD is weaker in adolescents due to fragility in self-consciousness and awareness [25,77]. In addition, individual vulnerabilities may also increase this trend. As the development of self-consciousness is closely linked with self-emotional development, SOD has been shown to be disturbed in some psychopathologies such as borderline personality disorder [78].(3)Collusion between adolescence and radicalization. Emotions and issues elicited by radicalization echo normal adolescent issues (such as guilt, shame, sexuality, and the need for rupture and change), just as the offering of radicalization resonates with adolescents’ usual coping mechanisms (such as projective identification, polarized attribution of values, intellectualization, and ascetism) [58,79], thus increasing self-other merging.(4)Adaptation of propaganda to target. We must acknowledge that recruiters have shown acute abilities in the cognitive aspects of empathy; these abilities are supported by recruiters’ ability to propose a wide range of propaganda messages [31] and then to subsequently adapt the message to each specific target [37]. This is not neutral for these young people; the perception of being the target of a perspective-taker have been shown to lead to an important self-other merging [80] as well as a soothing feeling of being understood [81]. This relief of no longer being alone to face uncertainty seems to play an important role in the early stages of the radicalization of young people [79].

Among adolescents, this process may favor bonding with one favorite person, often a charismatic leader [63]. Among young women who became especially affected by recent radicalization [28], this process may promote love fusion with a “charming prince” as frequent “sleeping beauty” storytelling suggests [37,82]. This privileged link then extends to the other members of the terrorist group; research has indeed robustly demonstrated that perspective-taking reduces stereotyping [83,84,85], with stereotype reduction extending to the whole target group [63,69,86]. This self-other overlap leads to acting stereotypically by adopting the behaviors of the target group [87], thus favoring a loss of personal identity, as highlighted among young radicalized people [88,89].

Taken together, these considerations support that the S/E system may be strongly involved in the process leading to radical group membership. In particular, these findings support the hypothesis that at least some of the young people who engage in radicalization would not be characterized by a lack of empathy per se; normal or even enhanced dispositional empathic capacities, and the interplay of internal and external factors, may, on the contrary, favor a sympathetic relationship with group members, thereby leading to self-identification and an altered sense of identity (or group contagion).

### 2.4. Motivational Aspects of SE and Radical Commitment

#### 2.4.1. Altruism

Another possible implication of empathy in radicalization relies on the importance of the motivational facets of empathy. Empathy includes moral values that involve caring for others’ well-being. This intuition is based on empathetic concern, which is the empathy subcomponent that reflects an other-oriented motivation that merges very early in development, i.e., earlier than the full acquisition of theory of mind and verbal abilities [90]. This intuition is the motivational part of what is called altruism.

Youth who engage in radicalization are often driven by humanitarian concerns rather than by violent radicalism [40,91,92]. Moreover, biographies of such youth often reveal pro-altruistic characteristics, such as the realization of humanitarian camps or vocations of medical and social careers, especially among girls [37]. These pro-altruistic characteristics do not always protect these youth from becoming violent extremists; i.e., these characteristics have often been identified and used by ISIS recruiters to select and convert potential violent extremists, particularly among adolescents and young adults [30]. Those who become involved can then give up everything, including family, friends, and usual activities, even when doing so is costly; i.e., becoming involved in violent extremism can lead to martyrdom, or what has been described as an altruistic suicidal behavior [8,93]. These findings show how, according to SE theory, high-level empathy profiles can favor violent actions rather than inhibit them [94].

Moreover, if empathy is commonly associated with unconditionally helping those in need, there is some evidence indicating that several interpersonal factors can interfere; these factors include how similar the target is to the observer [95] or how likable the target is. For example, empathy-related responses to pain in others are significantly reduced when one observes an unfair person receiving pain and when the observation co-occurs with increased activation in reward-related areas; the reduction in empathy-related responses to pain in others is correlated with an expressed desire for revenge [96]. This study underlines that empathy is shaped not only (at least in men) by the evaluation of other people’s social behavior but also by the promotion of the physical punishment of unfair opponents; this finding echoes evidence for altruistic punishment. A priori stigmatization of the target may also interfere with empathy, especially if the target is blamed: participants are more sensitive to the pain of targets infected with AIDS as a result of blood transfusion and less sensitive to pain caused by intravenous drug use. Importantly, the more the participants blamed these targets, the less pain the participants attributed to the targets [97].

These findings are all the more important if we consider that people who do not belong to the terrorist group are blamed and perceived as worthless and responsible for the threat [98]. In addition, empathy is also modulated by other factors in intergroup relations; these factors will be further specified.

#### 2.4.2. Group Belongingness

Group belongingness is central in adolescence when identity is at stake. In the context of radicalization, youth build new affiliation links in a surrogate family: they exchange names and identities with new sisters and brothers with whom they can find significance [99]. Importantly, brain regions [100] and neurochemical pathways (e.g., oxytocin, see [101]) involved in empathetic concern are similar to those implicated in parental and care-giving behavior selected through evolutionary perspectives across many species, supporting a major role of empathy in group belongingness [102], including in radicalization [103].

#### 2.4.3. Self-Regulatory Emotional Control

The failure to apply sufficient self-regulatory emotional control over the shared state leads to the experience of emotional contagion and, in the case of negative emotion, to personal distress (PD) [104]. PD is an aversive emotional reaction to the vicarious experience of another’s emotion; this experience results from perceiving another’s distress and is similar to the target’s state [105]. Importantly, the motivational behavioral response resulting from PD differs from empathetic concern since PD is a self-oriented motivated response [106] that has also been shown to be unrelated to prosocial behaviors [107] in adolescents [108].

Some factors may even foster PD upon empathetic concern, such as (i) psychological state of the observer (especially depression) [104] (depression is frequent among radicalized young people); (ii) exposure to physical rather than psychological pain [109]; and (iii) psychological inflexibility in adolescents’ prejudices [110]. Conversely, the adoption of radical views has been shown to reduce PD [111]; therefore, PD may have an important role as an acute coping mechanism possibly involved in the radical belief system.

PD may also mediate the relation between empathy and obedience. The famous Milgram paradigm is a paradigmatic form of dilemma eliciting empathy where people obey an instruction that involves harming another person [112]. Interestingly, using this obedience paradigm in ‘virtual’ life with virtual reality technology, Cheetham showed an atypical pattern of brain activity distinct from those commonly associated with affect sharing and compatible with PD [113]. These findings provide interesting insight into the way killing people may be perceived as the lesser of two evils when people are under threat (for example, compared to when pain is potentially inflicted on a person’s in-group relatives). The obedience paradigm supports the idea that inflicting pain on others may be seen as less important when it is committed under the pressure of an authority figure; this issue is easily transposable to radicalization, given that radicalization is generally influenced by a more or less legitimate religious authority [103].

#### 2.4.4. Perceived Injustice or Unfairness

Another determinant motivation for young radical commitment is unfairness [114,115]. Indeed, perceived injustice is an important determinant of radicalization in adolescence [98]. A triggering event that deals with injustice may become a determining factor in acting out [116]. These findings support that radicalized youth may react with stronger resentment when facing justice issues; such a reaction depends highly on the concept of justice sensitivity [117]. While some authors thought justice sensitivity to be linked with emotional components [118], some more recent studies indicate that it is more related to cognitive components of empathy [119]. Experiencing unjust events (directed against oneself or one’s community) can intensify justice sensitivity [120]. Interviews with young people who engage in radicalization often contain distressing events, such as emotional deficiencies, trauma, and abandonment [1,121,122]. Interestingly, the severity of past adversity, including adverse life events and childhood trauma [123], can lead to an increase in prosocial and altruistic behavior mediated by empathy [124]. Taken together, these findings suggest that empathy, including justice sensitivity, may be higher in radicalized young people. In addition, social neuroscience studies have shown that when participants are exposed to moral decision-making, cognitive empathy modulates functional connectivity across several domain-general systems, particularly in regions of the prefrontal cortex involved in goal representations in the service of moral decision-making [125]. These findings support that among those with higher empathy, perceiving injustice may provide a strong motivation to act to avoid injustice or restore justice. Generally, this research emphasizes the centrality of the motivational aspects of empathy in radicalization.

## 3. Consistency between Empathy and Radical Beliefs and Behaviors and the Type of Empathy Possibly Involved in Radicalization

### 3.1. Is Empathy Consistent with Radical Views, Beliefs and Behaviors?

Religious fundamentalism represents a distinctive attitude of certainty in the ultimate truth of one’s religious faith [126]. The relationship between religious fundamentalism and radicalization in youth is complex. Although some authors have downplayed the role of religious fundamentalism in radicalization [127], several scholars have noted the major role of religious fundamentalism in providing an identity figure [58,128]. However, in regard to strict fundamentalism, radicalization in youth is generally associated with radical views, beliefs and behaviors and can occur outside religion [129].

Although motivational aspects involved in radical views have been largely studied, little is known about the specificities of the cognitive processes and style of radicalized people. Several authors have suggested that cognitive inflexibility, defined as the inability to switch between modes of thinking and changing rules of categories, may predict extremist attitudes [130]. From cognitive functioning [131] to moral reasoning [132], the ability to entertain different perspectives is a crucial mechanism related to empathy. Notably, at least empathy’s cognitive aspects, such as perspective-taking and theory of mind, require the individual to hold in mind multiple perceived realities and active considerations of beliefs and views that differ from one’s own [133]. For these reasons, it might be natural to assume that these abilities may be decreased under fundamentalism and, more generally, negatively associated with radical thinking.

Importantly, the onset of a heterocentered perspective follows different stages of complexity during development [134]. The operational capacity of such a perspective is acquired between the ages of seven and ten, which is a critical period in the child’s development; during this period, the capacity of understanding, interpreting and accepting the plurality of viewpoints emerges [24]. This ability can be altered by neurodevelopmental disorders [135] and environmental factors, e.g., parents’ attention to their child’s mental states nurtures symbolic abilities [136]. Some authors noticed specificities in the family environment of youth who engage in radicalization, such as absence of countervailing opinions, lack of corrective answers to the subject’s radical position [137] and permissive arenas with little response to radical opinion [138]. When radicalization is underway, the isolation of the subject increases this trend even more [103].

Fundamentalism is also generally associated with violence, authoritarianism and aggression and has been negatively associated with empathy [139]. However, personality characteristics driving religious and nonreligious fundamentalism show a divergent relationship between fundamentalism and empathy, as religious fundamentalists scored higher for empathy than nonreligious ones [140]. In addition, empathy networks involve two brain networks that are anatomically dissociable and functionally antagonistic, namely, the task-positive network (TPN) also called the analytic network, and the default mode network (DMN), also referred to as the social brain. These two networks allow different types of thinking related to different belief considerations [141]. The thought process of people cycles between the two networks. In the religious fundamentalist’s mind, the DMN appears to dominate, while in the nonreligious fundamentalist’s mind, the TPN appears to rule [140]. As a consequence, empathic concern can be linked to hostility in individuals who focus on potential threats to protect what they regard as precious or sacred. Interestingly, the neural basis of sacred value showed activation in the left inferior frontal gyrus, a region associated with rule retrieval, opposite to utilitarian cost–benefit reasoning [142], and this brain region is also the core structure of emotional empathy involved in the MNS [143]. Collectively, these findings suggest that affective empathy may be higher in some radicalized people defined as “devoted actors” (as opposed to rational actors), who are particularly prone to making costly and extreme sacrifices in the defense of sacred values [70].

### 3.2. Disentangling Affective and Cognitive Empathy

Relationships between empathy and radicalization are complex, especially because most studies used multifaceted sympathy/empathy scales that do not enable disentangling the specific links that each component might have with radicalization.

Regarding the cognitive components of empathy, most research has focused on religious beliefs. Several authors have claimed that there is a positive association between mentalizing and belief in God, while considering that the capacity to perceive minds is not limited to human targets [144] and evidence of TOM network activation during prayer [145]. However, other authors found contradictory results [146]. Interestingly, these conflicting results may be related not to the degree but rather to different styles of belief: higher TOM may facilitate symbolic religious belief [147], while at the other extreme, TOM may be negatively related to a tendency to an “overliteral” understanding of language [148]. This overliteral understanding appears to be frequent among religiously radicalized youths [39], as has been previously observed in a population of alexithymic adolescents and young adults [149].

More generally, the literature mostly identifies a positive association between religiosity and empathy [150], cognitive flexibility and openness [151] and prosocial behaviors [152]. The theological account of the relationship between empathy and religion derives from the theory that religion generally promotes helping behavior and openness. Interestingly, some authors directly compared mentalizing measures to empathetic concern and found that the latter was a more robust and substantive correlate of religious belief [153].

Very recent research went further to clarify reciprocal relationships between empathy and religious beliefs. To determine whether holding religious beliefs promotes cognitive flexibility and openness or biases the development of such beliefs, Cristofori et al. studied neurological patients with brain lesions involving the main regions of TOM (notably the VmPFC) [111]. The authors tested two alternative hypotheses: whether empathy promotes religious belief or whether religious belief promotes greater prosocial tendencies. They concluded that, contrary to what was commonly admitted, empathy does not influence the development of religious beliefs, whereas religious cognition (relying on TOM) regulates empathetic responses to others. Interestingly, in this study, the image of God was the mediator between religious cognition and empathy. This echoes previous experiments that showed that among adolescents, an image of God as the “God of mercy” was associated with higher levels of empathy, while an image of God as the “God of justice” was associated with lower levels of empathy. Interestingly, in the same study, the image of God was associated with self-concept and self-esteem [154].

Importantly, if religious cognition modulates empathic abilities, there is serious concern about the potential deleterious effects of sustained radical thinking when shared religious beliefs no longer influence empathetic tendencies. This issue may be particularly important for young people who, between 2015 and 2018, fled France to join the war zones in Syria, as shown by the experts’ clinical observations of those returning from Syria in the qualitative study mentioned above [39].

## 4. Intermediary Summary

Let us summarize some key points: (i) empathy is a multifaceted construct, with affective and cognitive components both relying on different brain networks [13]; (ii) possible dissociation between cognitive and affective empathy has been shown in these networks in the context of lesion models [155] and psychopathology [156,157]; (iii) weaker TOM was associated with a greater God image thought that was associated with higher empathic concern [111]; (iv) lower perspective-taking abilities were associated with higher affective empathy among religious fundamentalists [140]; and (v) motivational components may modify the SE system [10,158].

Taken together, these findings support that in addition to radicalized youth possibly showing a deficit of empathy, other youth might present a dissociated profile combining a higher affective (including emotional) component of empathy and weaker cognitive components of empathy driven by motivational components. This can result in a strong adhesion of the emotional system to radical beliefs without adequate cognitive evaluation and criticism.

## 5. Radicalization and Empathy in an Intergroup Context

### 5.1. In-Group vs. Out-Group

The ontogenetic development of empathy shows that empathy constitutes an advanced skill that has been selected to enable thriving within a group context since group living offers several reproductive and long-term survival advantages compared to the advantages of going solo [103]. This suggests a double-edged feature of empathy when considering intergroup context: on the one hand, empathy enables strong cooperation, in-group efficiency and prosperity; on the other hand, when doing so, empathy implies that outsiders are excluded or harmed.

Numerous authors have supported the importance of in-group versus out-group polarization among macroenvironmental factors involved in radicalization (e.g., [98]). According to Social Identity Theory [159], people are inclined to perceive their group as better than most other groups and to develop an ‘us and them’ perspective as a consequence of this social categorization, leading to antagonism between groups. This categorization also maximizes intergroup differences; this maximization is not without consequences as far as empathy is concerned. Studies of empathy in an intergroup context indeed have shown that group membership can compromise all levels of empathic response (i.e., affective, cognitive, and motivational) and helping behavior [160]. Indeed, despite the common opinion that affective resonance is automatic, group membership can affect its induction in the observer. For instance, in a transcranial magnetic stimulation study, no vicarious mapping of the pain of individuals culturally marked as outgroup members on the basis of their skin color was found [161]. This reduction in emotional sharing in response to outgroup members also extends to emotional pain [162]. Importantly, in these studies, higher levels of racial prejudice were associated with a greater absence of empathetic response to outgroup members. In the same way, cognitive empathy is also modulated by group membership and can lead to biases that affect decision-making [163].

In-group favoritism also exists in group contexts other than ethnicity, for instance, in the context of a sports team [164], but also when group differences are generated artificially by using the minimal group paradigm according to minimal group theory [165]. Notably, Van Bavel showed that an arbitrary temporary novel group can override the effects of predominant group membership by inhibiting automatic racial biases in the context of mixed race teams [166]. As far as religious allegiance is concerned, empathetic response has been shown to be larger when participants viewed a painful event occurring to a hand labeled with their own religion than to a hand labeled with a different religion, and this classifier was generalized successfully to validation experiments in which the in-group condition was based on an arbitrary group assignment [167]. Interestingly, in this study, the neural empathetic response was modulated by minimally differentiating information (e.g., a simple text label indicating another’s religious belief).

A very important aspect of intergroup empathy bias is that such bias appears to depend highly on the social motivation of the perceiver. Indeed, various studies have shown that self-categorization along an in-group/out-group distinction is flexible and that recategorization with an arbitrarily defined group may be sufficient to overcome automatic response biases [168]. This is of special importance in the context of radicalization since young people recategorize their group of membership by establishing new kinship relations as an effect of propaganda [103].

Interestingly, social group membership is highly flexible and context-dependent, and not all outgroups elicit the same intergroup empathy bias. Cikara et al. highlighted two critical factors: functional relationships between groups (shared, competing, or independent goals) and relative group status [169]. These findings are of special relevance since the same conditions have been identified among macroenvironmental factors involved in group polarization in radicalization [89,129,170,171].

Previous research has shown that minimal group manipulation can enhance intergroup biases when groups are in competition. Notably, similar group membership between a helper and a target (whether the group is real or artificially determined) reinforced the role of empathy and helping thought. Simply categorizing participants into irrelevant social groups appears to be sufficient to facilitate an in-group bias in empathy for physical pain [172].

Importantly, intergroup empathy bias has been shown to be associated with enhanced hostility and pleasure in response to the out-group target’s misfortune (schadenfreude) [173]. In a neuroimaging study involving passionate fans of two baseball teams, in-group team failures were associated with increased activity in neural areas associated with the subjective experience of pain [174]. In contrast, out-group team failures were associated with increased self-reported pleasure and activity in neural areas associated with reward processing. Furthermore, the more positive value (pleasure) participants attached to the rival team’s failures, the more willing they were to assault a rival team fan. These results support the idea that outcomes of social group competition can directly affect primary reward-processing neural systems, with implications for intergroup harm [175].

Moreover, dissymmetry of status and resources even without overt competition can affect the intergroup empathy response [176]. These findings may be of special importance in the context of radicalization since, as shown by various authors working specifically with this population, dissymmetry of status and resources (real or symbolic) is an important macroenvironmental factor in hostility toward the out-group [40,98,170].

### 5.2. Empathy and Dehumanization

In regard to violent radicalization, dehumanization of the out-group is also of special importance. Surprisingly, several findings indicate that empathy processes may contribute to dehumanization against the out-group. Dehumanization is enabled by the formation of stereotypes, and research in social cognition firmly establishes that people differentiate each other not simply along an in-group vs. out-group boundary but also according to the extent to which they (dis)-like and (dis)-respect a target. In particular, the stereotype content model organizes beliefs about social groups along two fundamental dimensions: perceived warmth and competence [177].

Although numerous studies support the fact that empathy and specifically perspective-taking components can help form social bonds by decreasing prejudice and stereotyping [69,84,86], other findings indicate that perspective-taking can be a double-edged sword that also leads to exacerbated intergroup relations [178,179]. Indeed, assuming the perspective of a stereotype-consistent target may increase stereotyping when the target individual is highly stereotypic [180] and in the case of the perspective-taker’s need for cognitive closure [181]. Moreover, perspective-takers have been shown to be more likely than nonperspective-takers to adopt the negative stereotypical traits and behaviors of the person he or she is perspective-taking with [87].

Importantly, empathy can not only increase stereotyping but also be shaped by the stereotypes that result from it. Fiske examined social groups that elicit dehumanized emotions, such as disgust, and demonstrated that these individuals’ judgment is processed in a region anatomically distinct from that which is used in the case of social groups that elicit exclusively social emotions (such as pity, envy, and pride) [49]. These findings suggest that extreme out-groups do not elicit these complex social emotions in the perceiver and that the in-groups judge the out-groups as not experiencing complex emotions in the same way in which the in-group does [182]. These findings are compatible with infrahumanization theory [183], which states that people see some groups as less human than others and suggests that empathy may paradoxically contribute to this less human perception or dehumanization [184].

## 6. Discussion

To understand how empathy as a general concept (with its emotional, cognitive and social levels and its motivational and environmental influences) contributes to radicalization among adolescents and young adults, it is crucial to consider two contradictions.

### 6.1. The Contradiction of Empathy

First, several arguments seriously reduce the hegemonic strength of the commonly admitted empathy-altruism theory. Critically, this depends on the separateness of the self from the other. Without a distinct self and other and without distinct motivations to help the self or the other, it is impossible to detach altruism from egoism [185]. However, studying both sympathy and empathy provides interesting insight into how seemingly similar behaviors can be underpinned by different or even opposite phenomenological realities, especially in martyrdom cases [186]: while some pursue a more narcissistic motivation to die “in apotheosis” [187], others are driven by identity fusion and willingness to fight and die for the in-group [76].

Second, both sympathy and empathy, like social cognition in general, always depend on the context in which they occur [188]. Significantly, this implies that SE failures toward out-groups do not seem to depend on a person’s characteristics and that even the most deeply sympathetic person can mute his or her sympathy response to a perceived enemy under certain circumstances. This is especially true in situations such as radicalization, which involves several possible targets of the SE system (in-group members versus out-group members that can become potential victims).

Finally, the best predictor of meaningful intergroup attitudes and behaviors might not be the general capacity for sympathy but rather “parochial empathy” [189] or “intergroup empathy bias” [174], i.e., how empathy is distributed and especially the difference in sympathy felt for the in-group versus the out-group. In other words, the more sympathy one feels for one’s own group, the more likely one may be to endorse, support, or commit violence against the out-group and may lose the ability to feel what an out-group individual may feel (lack of empathy). This hypothesis has been confirmed by Bruneau, who found in three different intergroup contexts that out-group empathy inhibited intergroup harm and promoted intergroup helping, whereas in-group sympathy had the opposite effect. In all samples, in-group and out-group sympathy/empathy had independent, significant, and opposite effects on intergroup outcomes, thus controlling trait empathic concern [189].

Thus, at the end of this review, we hypothesize a double dissociation model of empathy in radicalization:

The first dissociation occurs at the individual level and is characterized by high affective SE contrasting with diminished cognitive empathy. This dissociation can lead to a strong identification with the radical group and adherence to a radical belief system due to the absence of adequate emotional regulation and cognitive criticism mechanisms.

The second occurs at the intergroup level, thus combining high in-group empathy and low out-group empathy, both inversely related according to a parochial empathy effect; i.e., an increase in one leads to a decrease in the other.

This double dissociation model is summarized in Figure 1 from a micro-macro perspective, as proposed in Campelo et al.’s model of adolescent radicalization [1]. Importantly, this model has both theoretical and very concrete applications; i.e., this model can explain why programs aimed at increasing empathy to prevent radicalization may have mixed results or even paradoxical effects [190]. In particular, Feddes et al. found unwanted harmful effects, as participants showed increased narcissistic traits (an identified risk factor for radicalization) after the implementation of this type of program among adolescents [191]. Consequently, although it still needs to be confirmed systematically, this model may be a first step toward practical implications of the SE system on countering violent extremism intervention programs.

### 6.2. The Contradiction of Radicalization among Adolescents

The issue of preserved or altered SE systems seems to be quite paradigmatic of a more general apparent contradiction concerning the complex relationships between radicalization and young people. Indeed, the same contradictory results are found in the literature regarding the prevalence of mental disorders among radicalized adolescents and young adults: while several authors emphasize their rarity and the lack of any consistent psychiatric or psychological profile, other studies consider that the situation is less contrasted and show an overrepresentation of mental disorders in the radicalized young [39,192,193]. This contradiction is probably due to the imprecise definition of radicalization: which motive do we refer to when we talk about radicalization? Is there a radical claim? Is there an intention? Does radicalization mean acting out? This is a crucial issue because while radicalization may be a “new symptom” related to the complex issues of adolescence in some cases [79], serious acts may also be the prerogative of another youth typology that differs from the first one.

Significantly, forensic reports of youth involved in serious terrorist acts also find characteristic unemotional traits with real sympathy deficits in which ideology is used as violence. Some authors even underlined the observation that the radicalized individuals who committed radicalized murders were all engaged in severe delinquent behaviors when they were adolescents [121], especially in France [194], thus raising the possibility that these individuals should be considered as a particular subgroup needing to be specifically studied to shed some more clinical light on their psychopathological profile. However, a comparison between minors convicted in France for ‘criminal association to commit terrorism’ and teenagers convicted for nonterrorist delinquency shows that adolescents engaged in radicalization and terrorism do not have a significant prevalence of psychiatric disorders, suicidal tendencies or lack of SE. In addition, radicalized adolescents show better intellectual skills, insight capacities, coping strategies and less history of nonterrorist delinquent acts prior radicalization [195].

Those who join the ranks of terrorists to become warriors may constitute a third subgroup on the empathy continuum. If an SE deficit is not primitive in some radicalized individuals, it may perhaps emerge as the radicalization unfolds through a desensitization mechanism or a temporary dissociation. Some insight into this temporary dissociation in the SE system can be gained by exploring recent works about Syndrome E [196]. In particular, some authors have stressed the importance of distinguishing between the rule system and the brain valuation system which do not rely on the same brain networks. Obsessive compliance with rituals and procedures, such as those observed in radicalization, may play a major role in the rule system overriding the BVS [197]. In other cases, BVS may also be completely deviated by cognitive distortions amplified by transmission and social contagion in the belief, for example, that killing heretics can get one into heaven [56]. To favor violent engagement, recruiters have indeed well understood the power of dehumanizing out-group members to help recruits move to violent radicalization. Adolescents may be more concerned with this rule system overriding the BVS, considering the immaturity of their decision-making system [198]. This hypothesis however remains to be confirmed.

Moreover, the SE system is not a reliable source of information in moral decision-making as this system is unconsciously and rapidly modulated by various social signals and situational/motivational factors [10]. In the case of adolescent radicalization, we individuate at least three different contributions related to three possible trajectories of young radicalized individuals according to their SE status at baseline. In Figure 2, we describe the SE trajectories related to these three subgroups.

**Figure 2 children-09-01889-f002:**
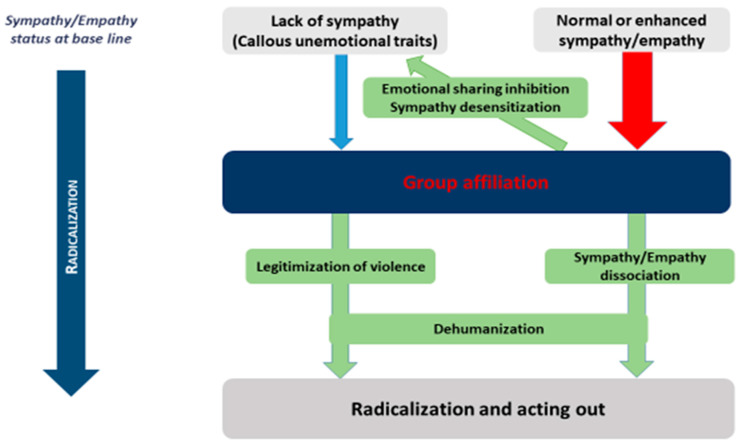
The sympathy/empathy trajectories and changes during radicalization according to the youth empathy status at the baseline. The model shows 3 trajectories in green. Among individuals with a lack of sympathy (callous unemotional traits, left), legitimization of violence is enough to engage in radicalization and violence. Concerning individuals with normal or enhanced sympathy/empathy, we hypothesize two trajectories. First, based on sympathy inhibition/desensitization, the individual joins youth who lack empathy. The second trajectory is based on a sympathy/empathy dissociation (that combines ↗ sympathy for the in-group and ↘ cognitive empathy toward the out-group) favoring violence against the out-group.

The first group includes individuals with SE deficits or CUTs. In this group, legitimization of violence would be sufficient to move toward radicalization and possibly violent acts.

For individuals with normal SE, we hypothesize two trajectories: First, based on SE sympathy inhibition/desensitization, the individual joins youths who lack empathy. Second, based on an SE dissociation combining an increase in emotional sympathy for the in-group and a decrease in cognitive empathy for the out-group, the individual favors violence toward the out-group. 

This hypothesis requires further work to be demonstrated notably by assessing empathetic functioning of young individuals at different levels of the radicalization process and among different sub-groups. Especially, comparison could be made between empathic profile of non-terrorist habitual offenders, terrorist offenders, and actual terrorist perpetrators. This assessment may go further Bronsard’s study [195] by using a multidimensional scale, such as the Interpersonal Reactivity Index (IRI), which would allow for separate assessment of cognitive and affective components of empathy. More specifically, the search for correlations between, one the one hand, the different components of empathy and, on the other hand, the loss of SOD (such as that assessed by Sheikh et al. [70]) might confirm that increased affective contrasting with decreased cognitive empathic abilities may be involved in identity fusion with terrorists groups.

Furthermore, given the previously demonstrated relationships between empathetic profiles and personality traits [199,200,201] as well as psychopathological alterations [202], correlational analyses may be usefully conducted to examine whether some psychopathological categories highlighted in the literature about radicalization might reflect some dominance of one SE profile over the other. This may be particularly useful in some extreme cases, such as that studied by Merari [7] who compared psychological profiles of suicide bombers with terrorists imprisoned for unrelated offences. They found in the first group more avoidant-dependent personality disorders (60% vs. 17%), fewer psychopathic tendencies (0% vs. 25%) as well as fewer impulsive and unstable tendencies (27% vs. 67%).

Finally, without claiming to find a strict concordance between both neurocognitive and psychopathological profiles, we may assume that, following the three SE trajectories shown in Figure 2: (i) youth with low cognitive and affective developmental empathy may be predominant among youth with psychopathic traits (engaging in chronic antisocial behavior); (ii) youth with high affective empathy and low cognitive empathy in youth with borderline/paranoid functioning (with high inhibition of sympathy referring) (iii) youth with increased sympathy for in-group and reduced cognitive empathy for out-group may be predominant in youth with severe narcissistic vulnerability with dependency traits.

## 7. Conclusions

The conclusion of this review supports the hypothesis that young radicalized people may have specificities in the development of SE abilities leading to an atypical SE profile that differs from that of young people with other nonprosocial behaviors.

We claim that, far from being rooted in a total deficit in SE, radicalization may be related to a paradox combining normal and even enhanced empathy promoted by the accentuation of social identification with the in-group as opposed to poor empathy for the out-group. More importantly, preserved and even enhanced in-group empathic abilities among these individuals may play a crucial role in the violence directed at an out-group. Moreover, a hypothetical dissociation between higher affective versus lower cognitive empathic abilities may favor an overload of the emotional system with no compensation of either top-down regulation mechanisms or cognitive criticism. A transitory dissociation of empathy leading to radical acting out may occur, but such a dissociation remains scarce in youth. It should nevertheless be underlined that this profile may apply only to some of the adolescents who engage in radicalization and should not lead to the trivialization of this issue or to exclusion of the fact that some of them may indeed present a frank empathy deficit.

Finally, SE development suggests differences among evolutionary trajectories that may lead to radicalization. This is an essential prerequisite for a better specification of the clinical and neuropsychological underpinnings of radicalization, not only to detect but also to develop, at both the individual and collective levels, the appropriate responses.

## Figures and Tables

**Figure 1 children-09-01889-f001:**
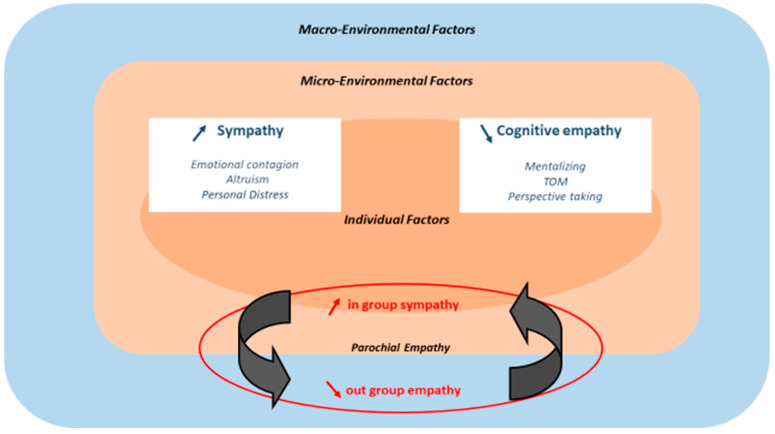
The double dissociation model of sympathy/empathy in youth radicalization. The model shows how sympathy/empathy may change during radicalization with a dissociative mode combining increased sympathy (emotional sharing and affective contagion) and decreased empathy (the cognitive ability to feel what the other feels) according to the sympathy/empathy status at the baseline and specific trajectories summarized.

## Data Availability

Not applicable.

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
