# Peer review of "Sympathy-Empathy and the Radicalization of Young People"

_children, 2022, doi:10.3390/children9121889_

Round 1
Reviewer 1 Report
This article presents a very interesting and original take on radicalisation and the reasons for it. I very much enjoyed reading it and understanding more about empathy and radicalisation to terrorism. I believe it should be published, but I have a few comments that I would recommend be taken into account it is finally accepted for publication.
* The article begins with the problem that ‘young people are increasingly concerned’ [39] with radicalisation. Obviously the radicalisation of young people is still a concern, but I do feel that this framing reads a little more from around 2014/15, when increasing numbers of very young people (down to 12-15) were being drawn to ISIS, and either travelling overseas to join the organisation or commit acts of terror in their own countries. A lot of the radicalisation literature around foreign fighters focused on this problem. The authors mention the radicalisation of ‘young people without Muslim backgrounds in European countries’, which I assume refers to the growing threat of right-wing extremism, though they are not explicit about it. This shows an awareness of the changing landscape, and yet, with the growing threat of RWE, we are actually seeing a trend towards older (white) males radicalising. So, while understanding the radicalisation of young people will always be important, I am left a little unclear as to the ‘hook’ for the article’s focus on young people. I would suggest saying something around the radicalisation of young people was a key issue with ISIS/foreign fighters, and there is still a lot we don’t know about that phenomenon despite significant research. Now, while the threat of RWE is growing and shows some evidence of a tendency towards older rather than younger recruits, we know there are no clear patterns to radicalisation and so we still need to understand the radicalisation of young people as this will remain an ongoing threat and social challenge. That is not a big change to the wording, but something along those lines would help frame the issue more clearly.
* The initial framing issue is an easy fix, but it actually raises a more significant question for me: why the apparent focus on young people? The focus is easy enough to justify, so I mean this in the following sense: what the article contributes around sympathy/empathy and radicalisation is not unique to young people; the article provides a general review of the literature and its relevance to radicalisation processes and causes, almost regardless of age. I understand the authors may be adding the youth lens due to the journal’s focus, but I don’t think the editors would mind if there was some comment added in the introduction around this having more general relevance to radicalisation literature. To me, a clearer justification would be to say something like: while the sympathy/empathy model has relevance regardless of age, it is particularly important to consider this with young people as they are progressing through important stages of development and socialisation and this enhances the potential for protective measures to have some concrete effect. The authors already have some language in the introduction to this effect, I just think it would benefit from some more explicit comment around generalisability, especially given the context (mentioned in the comment above) that RWE radicalisation tends towards older individuals.
* In the introduction, background is provided on the sympathy/empathy model. Bearing in mind that I am not a psychologist, I have two questions here. First, why does sympathy drop off in favour of empathy as the core focus? It seems to me that the article could just be about empathy and radicalisation, rather than the sympathy/empathy model per se, as the balance is heavily skewed that way. Second, what is the importance of sympathy-empathy as a ‘model’ or ‘system’? I understand the relevance of empathy and I think this is a very important contribution to the literature; I haven’t read anything quite like it before. And yet, I am struggling to see the benefit of understanding sympathy-empathy as a ‘system’ or ‘model’, especially given that sympathy is not explained to the same degree as empathy. My suggestion would be to cut out some of the more detailed explanation from the introduction, and include the following in a revised Section 1: empathy, sympathy, the sympathy-empathy 'system' or 'model', and (briefly) the relevance of this model to radicalisation, including any existing literature. The model could also be related here to more established findings from the radicalisation literature (e.g. the lack of a clear sense of identity in young people - how does this relate?). I would then start in Part 2 on the more specific questions of empathy/sympathy and radicalisation. This would give more space to provide the necessary background on the SE model, rather than only including this in the introduction. Also, if it turns out that the authors do want to focus more on empathy rather than the ‘sympathy-empathy system’, then I would just frame the article and this Section 1 in those simpler terms. This may make the article’s key contribution (as I read it, around understanding empathy, terrorism, and the apparent paradox therein) clearer.
* Regarding the empathy ‘paradox’, I’m not sure I would call it a paradox, but I do think this is a very important research problem and apparent contradiction. It could be framed clearly from the outset as the article’s main ‘hook’. This might actually provide a clearer opening statement than the current justifications around age and the radicalisation of young people. The apparent contradiction, to put it a little crudely, is that terrorists kill civilians in large numbers (suggesting a clear lack of empathy) and yet their identification with a cause, the underlying feelings of injustice and grievance driving their radicalisation, and other ways they can help communities, suggests they may have actually have a lot of empathy (even an excess of it). I would call this a very interesting research problem, or an apparent contradiction - not necessarily a paradox - but it is a very interesting, original, and engaging way to frame the problem. I would suggest framing this problem clearly in everyday language in the abstract and introduction, rather than focusing on more technical ideas around the benefits of looking at the SE system.
* If the authors take into account these comments, adjusting the framing of the introduction and restructuring a clearer initial Section 1 (more generally on the SE system) before addressing specific radicalisation issues, I believe it would make an important contribution to the literature.
Some other minor points that may help in revising the article:
* The title could definitely be shortened, e.g.: ‘Sympathy-Empathy and the Radicalisation of Young People’ (?)
* The authors raise the question: ‘why does someone radicalise against his own people’? [42-3]. I’m not sure exactly what they mean by this, but given the mention of non-Muslim terrorism I take it as referring to the problem of homegrown right-wing radicalisation. The issue of who is a subject's 'own people' is actually quite complex, and it would be interesting to discuss this in relation to empathy. Who is a person’s ‘own people’ will differ depending on who they perceive as their in-group and who they perceive as an outgroup threat. This is regardless of the larger society or country they are all living in. For example, an act of homegrown right-wing extremism (such as an attack against worshippers at a mosque) would not be against the person’s ‘own people’, even if they all live in the same country. It would be very much against a perceived outgroup threat from Muslims, in line with the attacker's ideology. Indeed, they are very much likely to be perceived as 'not one of us', not people who belong in our country, etc. I am not at all saying these perceptions are justified, just trying to show that who is someone's 'own people' is a crucial and complex question with terrorist ideology. (Also, less crucially, I would keep the language here gender neutral - avoiding ‘his’ own people - even though statistically that is more accurate.)
* A few times throughout, there is mention of individuals ‘temporarily joining youths’. I’m not sure if the authors mean to say groups, as people can join and then leave groups over time, or whether the joining refers a coming together of individuals in a relationship, which can be fluid. Some greater clarity around this language would assist readers.
* The double dissociation model comes right at the end of the article (608). Arguably, this should be framed from the start as the article’s main contribution.
* I would recommend against putting a new table in the conclusion - I would move this up to where the double dissociation model is introduced, and (as above) frame that contribution more clearly from the start.
* The SE model is discussed in relation to face-to-face encounters in group situations, or online. The implication seems to be that these are two different things, but much of the research around online radicalisation suggests that terrorism is more likely to result when these two processes happen simultaneously (i.e. there is a lot of myth and media coverage around lone wolf radicalisation online, when typically people radicalise through a combination of online and offline radicalisation). This doesn’t change the article’s analysis at all, but it does make me think that SE has more to do with a personality trait or individual capacity, which is common to the subject, rather than playing out differently in online or off-line environments.
* There is mention of youths who ‘sometimes engage in radicalisation’ (155) - I am not sure about this, as radicalisation is a psychological process and not a behaviour that one can (choose to) dip in and out of. An individual may well go in and out of the process, stall, go backwards, then forwards, towards some eventual act of violence (or not) - but this is all part of the complex process we call radicalisation. ‘Sometimes engaging’ with radicalisation makes it sound more like a conscious choice, that involves engaging or not engaging with it at different times. It is a difficult question to say when somebody is *no longer* radicalising, but to me this language describes more accurately people joining and leaving terrorist groups, or engaging in criminal activity and then disengaging from it, rather than engaging or disengaging with radicalisation as a process.
Author Response
This article presents a very interesting and original take on radicalisation and the reasons for it. I very much enjoyed reading it and understanding more about empathy and radicalisation to terrorism. I believe it should be published, but I have a few comments that I would recommend be taken into account it is finally accepted for publication.
The authors are very grateful to the reviewer for his careful review of this article and for his very helpful suggestions for improvement.
* The article begins with the problem that ‘young people are increasingly concerned’ [39] with radicalisation. Obviously the radicalisation of young people is still a concern, but I do feel that this framing reads a little more from around 2014/15, when increasing numbers of very young people (down to 12-15) were being drawn to ISIS, and either travelling overseas to join the organisation or commit acts of terror in their own countries. A lot of the radicalisation literature around foreign fighters focused on this problem.
Thank you very much for raising this question which allows us to be more precise. The research discussed in this article has focused particularly on the literature dealing with the problems raised by Islamic radicalization among youth. In some countries, notably France, there has been an increase in the number of young people whose Islamic radicalization has developed in situ, raising similar problems in terms of security. This article is indeed based on the population of youth involved on the dates you indicated and this precision of the years 2014 /2015 has been added. However we do not have reliable evidences to say that this increase is still current for this radicalization. We thank you for bringing this to our attention and have therefore removed the adverb "increasingly".
The authors mention the radicalisation of ‘young people without Muslim backgrounds in European countries’, which I assume refers to the growing threat of right-wing extremism, though they are not explicit about it.
Thank you for revealing to us this ambiguity in our text. The non-Muslim youth we have mentioned are those who, despite their lack of Muslim background, have converted to radical Islam. Additionally, our work is based solely on Islamic radicalization, which is so far the most worrying and widespread issue in France. While right wing followers are present in France, Right-Wing Extremists are still rare in our country (much rarer than in Germany or Scandinavia, for example). They are not directly addressed either in local mental health research or in the clinical work with adolescents and young adults. This has been better specified in the introduction of our paper.
This shows an awareness of the changing landscape, and yet, with the growing threat of RWE, we are actually seeing a trend towards older (white) males radicalising. So, while understanding the radicalisation of young people will always be important, I am left a little unclear as to the ‘hook’ for the article’s focus on young people. I would suggest saying something around the radicalisation of young people was a key issue with ISIS/foreign fighters, and there is still a lot we don’t know about that phenomenon despite significant research. Now, while the threat of RWE is growing and shows some evidence of a tendency towards older rather than younger recruits, we know there are no clear patterns to radicalisation and so we still need to understand the radicalisation of young people as this will remain an ongoing threat and social challenge. That is not a big change to the wording, but something along those lines would help frame the issue more clearly.
The authors thank the reviewer for the comment.
We agree that what we present in this article should be considered for older radicalized individuals and other radicalization topics and have added sentences in the manuscript to acknowledge this.
In addition, our study does not allow us to conclude that the SE system we described here is specific to radicalization or whether it could apply to other forms of violent ideological sectarism with violence. Our Figure 2 shows that the early stage of radicalization may have common aspects with other types of non-violent ideological affiliation.
* The initial framing issue is an easy fix, but it actually raises a more significant question for me: why the apparent focus on young people? The focus is easy enough to justify, so I mean this in the following sense: what the article contributes around sympathy/empathy and radicalisation is not unique to young people; the article provides a general review of the literature and its relevance to radicalisation processes and causes, almost regardless of age. I understand the authors may be adding the youth lens due to the journal’s focus, but I don’t think the editors would mind if there was some comment added in the introduction around this having more general relevance to radicalisation literature. To me, a clearer justification would be to say something like: while the sympathy/empathy model has relevance regardless of age, it is particularly important to consider this with young people as they are progressing through important stages of development and socialisation and this enhances the potential for protective measures to have some concrete effect. The authors already have some language in the introduction to this effect, I just think it would benefit from some more explicit comment around generalisability, especially given the context (mentioned in the comment above) that RWE radicalisation tends towards older individuals.
We thank very much the reviewer for his important comment and his kind proposal of a formulation to take it into account. We have modified our text accordingly
* In the introduction, background is provided on the sympathy/empathy model. Bearing in mind that I am not a psychologist, I have two questions here. First, why does sympathy drop off in favour of empathy as the core focus? It seems to me that the article could just be about empathy and radicalisation, rather than the sympathy/empathy model per se, as the balance is heavily skewed that way. Second, what is the importance of sympathy-empathy as a ‘model’ or ‘system’? I understand the relevance of empathy and I think this is a very important contribution to the literature; I haven’t read anything quite like it before. And yet, I am struggling to see the benefit of understanding sympathy-empathy as a ‘system’ or ‘model’, especially given that sympathy is not explained to the same degree as empathy. My suggestion would be to cut out some of the more detailed explanation from the introduction, and include the following in a revised Section 1: empathy, sympathy, the sympathy-empathy 'system' or 'model', and (briefly) the relevance of this model to radicalisation, including any existing literature. The model could also be related here to more established findings from the radicalisation literature (e.g. the lack of a clear sense of identity in young people - how does this relate?). I would then start in Part 2 on the more specific questions of empathy/sympathy and radicalisation. This would give more space to provide the necessary background on the SE model, rather than only including this in the introduction. Also, if it turns out that the authors do want to focus more on empathy rather than the ‘sympathy-empathy system’, then I would just frame the article and this Section 1 in those simpler terms. This may make the article’s key contribution (as I read it, around understanding empathy, terrorism, and the apparent paradox therein) clearer.
We thank the reviewer for his important comment and his very helpful suggestion to improve the structuring and framing of the article. We have modified the article accordingly:
The introduction and the beginning of the abstract now focus on the main issue, which is the apparent contradiction between empathy and radicalization.
We have removed from the introduction all the elements concerning the presentation of the Sympathy/Empathy system to devote a specific part to it at the beginning of section 1 (section 1. 1). This sub-section makes it possible to clarify why we think it is useful to differentiate sympathy and empathy within the S/E system, taking into account the fact that they are often unduly confused.
* Regarding the empathy ‘paradox’, I’m not sure I would call it a paradox, but I do think this is a very important research problem and apparent contradiction.
We thank the reviewer for this comment and have replaced the term paradox with the term contradiction.
It could be framed clearly from the outset as the article’s main ‘hook’. This might actually provide a clearer opening statement than the current justifications around age and the radicalisation of young people. The apparent contradiction, to put it a little crudely, is that terrorists kill civilians in large numbers (suggesting a clear lack of empathy) and yet their identification with a cause, the underlying feelings of injustice and grievance driving their radicalisation, and other ways they can help communities, suggests they may have actually have a lot of empathy (even an excess of it). I would call this a very interesting research problem, or an apparent contradiction - not necessarily a paradox - but it is a very interesting, original, and engaging way to frame the problem. I would suggest framing this problem clearly in everyday language in the abstract and introduction, rather than focusing on more technical ideas around the benefits of looking at the SE system.
Thank you very much indeed for this very helpful suggestion; We reviewed the writing of the manuscript to take into account so that the apparent contradiction between empathy and radicalization appears as clearly as possible in the abstract and introduction as the main "hook" of the article, and provide a clearer opening statement.
* If the authors take into account these comments, adjusting the framing of the introduction and restructuring a clearer initial Section 1 (more generally on the SE system) before addressing specific radicalisation issues, I believe it would make an important contribution to the literature.
The authors thank the reviewer for his suggestions for improvement. They have taken the comments into account, adjusting the framing of the introduction and restructuring section 1 before addressing the specific issues of radicalization.
Some other minor points that may help in revising the article:
* The title could definitely be shortened, e.g.: ‘Sympathy-Empathy and the Radicalization of Young People’ (?)
We thank the reviewer for his very helpful proposal. We have simplified the title of the paper accordingly.
* The authors raise the question: ‘why does someone radicalize against his own people’? [42-3]. I’m not sure exactly what they mean by this, but given the mention of non-Muslim terrorism I take it as referring to the problem of homegrown right-wing radicalization.
We clarified what we wanted to mean by the ambiguous reference to radicalized individual without muslin backgrounds in the text of the paper.
The issue of who is a subject's 'own people' is actually quite complex, and it would be interesting to discuss this in relation to empathy. Who is a person’s ‘own people’ will differ depending on who they perceive as their in-group and who they perceive as an outgroup threat. This is regardless of the larger society or country they are all living in. For example, an act of homegrown right-wing extremism (such as an attack against worshippers at a mosque) would not be against the person’s ‘own people’, even if they all live in the same country. It would be very much against a perceived outgroup threat from Muslims, in line with the attacker's ideology. Indeed, they are very much likely to be perceived as 'not one of us', not people who belong in our country, etc. I am not at all saying these perceptions are justified, just trying to show that who is someone's 'own people' is a crucial and complex question with terrorist ideology. (Also, less crucially, I would keep the language here gender neutral - avoiding ‘his’ own people - even though statistically that is more accurate.)
Thank you for this comment. While we will clearly state that our paper is not addressing the issue of Right Wing Extremists for the reasons already mentioned we share the content of this comment and integrated the core of this reflexion in the paper.
* A few times throughout, there is mention of individuals ‘temporarily joining youths’. I’m not sure if the authors mean to say groups, as people can join and then leave groups over time, or whether the joining refers a coming together of individuals in a relationship, which can be fluid. Some greater clarity around this language would assist readers.
We thank you the reviewer for his comment and clarified the language accordingly
* The double dissociation model comes right at the end of the article (608). Arguably, this should be framed from the start as the article’s main contribution.
As recommended, the double dissociation model is presented at the beginning of the article, at the end of the introduction, as the main contribution of the article.
* I would recommend against putting a new table in the conclusion - I would move this up to where the double dissociation model is introduced, and (as above) frame that contribution more clearly from the start.
Following the above, the figure summarizing the double dissociation model of empathy (Fig 1) is presented much earlier, at the same time as this hypothesis is introduced, in order to more clearly manifest this important contribution of the article from the very beginning.
* The SE model is discussed in relation to face-to-face encounters in group situations, or online. The implication seems to be that these are two different things, but much of the research around online radicalisation suggests that terrorism is more likely to result when these two processes happen simultaneously (i.e. there is a lot of myth and media coverage around lone wolf radicalisation online, when typically people radicalise through a combination of online and offline radicalisation). This doesn’t change the article’s analysis at all, but it does make me think that SE has more to do with a personality trait or individual capacity, which is common to the subject, rather than playing out differently in online or off-line environments.
The SE model is indeed discussed in relation to face-to-face encounters in group situations, or online.We are in complete agreement with the reviewer that terrorism is more likely to occur when these two processes occur simultaneously. We revised the paper to make it clearer.
The aim of individualizing these different processes was to be as comprehensive as possible rather than to separate or contrast them. In particular, the important point was to highlight that the same empathic capacities (which are indeed intrinsic to the subject) are not only solicited in real face-to-face exposures, but also in virtual exposure situations.
* There is mention of youths who ‘sometimes engage in radicalisation’ (155) - I am not sure about this, as radicalisation is a psychological process and not a behaviour that one can (choose to) dip in and out of. An individual may well go in and out of the process, stall, go backwards, then forwards, towards some eventual act of violence (or not) - but this is all part of the complex process we call radicalisation. ‘Sometimes engaging’ with radicalisation makes it sound more like a conscious choice, that involves engaging or not engaging with it at different times. It is a difficult question to say when somebody is *no longer* radicalising, but to me this language describes more accurately people joining and leaving terrorist groups, or engaging in criminal activity and then disengaging from it, rather than engaging or disengaging with radicalisation as a process.
Thank you for this very important comment. Our experience shows that while some clinical cases show that some young people engage and then disengage in radicalized views, values, and beliefs, this is clearly not the case for all radicalized youth. We have clarified the language we have used in this paper to ensure that it will not be misunderstood in this regard.
Reviewer 2 Report
Thank you for your interest in this topic. Your manuscript has important implications for how we understand radicalization and the relationship to SE. I would have liked to see an address of Social Identity Theory in your discussion of in-group out-group concepts, but otherwise I felt this was a thorough review of the literature.
Author Response
Thank you for your interest in this topic. Your manuscript has important implications for how we understand radicalization and the relationship to SE. I would have liked to see an address of Social Identity Theory in your discussion of in-group out-group concepts, but otherwise I felt this was a thorough review of the literature.
We thank the reviewer for his very careful proofreading and positive comments.
Although Social Identity Theory was very present in the article, especially in understanding the concepts of in-group and out-group, a really explicit reference to this theory was missing. We thank the reviewer for allowing us to correct this with the addition of the appropriate bibliographic reference (159).
Reviewer 3 Report
Attached is a sheet with the few typos I found. Other than that, the only main comment I had was the suggestion that you note the age range that you are dealing with in the beginning of your submission. As the Journal deals with children and adolescents, it would help focus the reader from the beginning. Are you looking at 12+, 13+ or from when to when primarily? Other than that, I appreciated the resources you introduced.

Author Response
Attached is a sheet with the few typos I found. Other than that, the only main comment I had was the suggestion that you note the age range that you are dealing with in the beginning of your submission. As the Journal deals with children and adolescents, it would help focus the reader from the beginning. Are you looking at 12+, 13+ or from when to when primarily? Other than that, I appreciated the resources you introduced.
The authors would like to thank the reviewer for his careful proofreading, for correcting the typos and for raising the question concerning the age group concerned by the article.
In terms of the age group targeted, it is true that the framework of this article is very concerned by the growing number of very young people, as young as 12/13 years old (this age limit has been added to the manuscript), who have been attracted to ISIS and have traveled (or tried to travel) abroad to join the organization or commit acts of terrorism in their own country in the period during which adolescent and young adults were particularly at risk of Islamic radicalization (2014-2018)
As the reviewer rightly pointed out, we did not specify a particular age range in our article. We believe indeed that the S/E model we present is relevant regardless of age. We clarified it in the manuscript.